# Innate Immune System in the Context of Radiation Therapy for Cancer

**DOI:** 10.3390/cancers15153972

**Published:** 2023-08-04

**Authors:** Ettickan Boopathi, Robert B. Den, Chellappagounder Thangavel

**Affiliations:** 1Center for Translational Medicine, Department of Medicine, Thomas Jefferson University, Philadelphia, PA 19107, USA; 2Department of Radiation Oncology, Sidney Kimmel Cancer Center, Thomas Jefferson University, Philadelphia, PA 19107, USA; robert.den@jefferson.edu

**Keywords:** radiation therapy, DNA damage, dendric cells, NK cells, macrophages, hypoxia, tumor microenvironment, microbubble oxygen

## Abstract

**Simple Summary:**

Ionizing radiation therapy is an important component of cancer treatment. This review provides a summary of the latest advancements, clinical use, and limitations of radiation therapy. Moreover, this review explores how radiation affects the body’s natural defense system, which plays a crucial role in fighting cancer. The immune responses triggered by radiation therapy help the body eliminate tumors naturally. We explore how radiation therapy triggers the immune cells of the body, like macrophages, dendric cells, and natural killer cells, to kill tumor cells effectively. The review also offers strategies to enhance the effectiveness of radiation therapy while preserving the body’s innate immune system. This comprehensive review is valuable for researchers, as it sheds light on the cutting-edge developments in radiation therapy and its potential impact on cancer treatment. This newfound knowledge can ultimately lead to improved cancer treatments and better outcomes for cancer patients.

**Abstract:**

Radiation therapy (RT) remains an integral component of modern oncology care, with most cancer patients receiving radiation as a part of their treatment plan. The main goal of ionizing RT is to control the local tumor burden by inducing DNA damage and apoptosis within the tumor cells. The advancement in RT, including intensity-modulated RT (IMRT), stereotactic body RT (SBRT), image-guided RT, and proton therapy, have increased the efficacy of RT, equipping clinicians with techniques to ensure precise and safe administration of radiation doses to tumor cells. In this review, we present the technological advancement in various types of RT methods and highlight their clinical utility and associated limitations. This review provides insights into how RT modulates innate immune signaling and the key players involved in modulating innate immune responses, which have not been well documented earlier. Apoptosis of cancer cells following RT triggers immune systems that contribute to the eradication of tumors through innate and adoptive immunity. The innate immune system consists of various cell types, including macrophages, dendritic cells, and natural killer cells, which serve as key mediators of innate immunity in response to RT. This review will concentrate on the significance of the innate myeloid and lymphoid lineages in anti-tumorigenic processes triggered by RT. Furthermore, we will explore essential strategies to enhance RT efficacy. This review can serve as a platform for researchers to comprehend the clinical application and limitations of various RT methods and provides insights into how RT modulates innate immune signaling.

## 1. Introduction

Cancer remains the leading cause of death globally. According to the American Cancer Society (ACS), in 2023, an estimated 1,958,310 new cancer cases and 609,820 deaths from all cancers are anticipated to occur in the United States [1]. Cancer incidences increased for prostate cancer (PCa) by 3% annually from the year 2014 through 2019, following two decades of decline, translating to an additional 99,000 new cancer cases [1]. RT is one of the most effective forms of oncology care and continues to play a major role in the treatment of various types of malignancies. The emergence of advanced technologies has propelled a revolution in the field of RT, empowering clinicians with tools and techniques to deliver accurate and safe delivery of radiation doses to tumor cells. The advancement in intensity-modulated radiation therapy (IMRT), stereotactic body radiation therapy (SBRT) and proton therapy have revolutionized the field. These technological advancements have resulted in increasingly conformal radiation treatments. In this review, we present the technological advancement in various types of RT methods and highlight their clinical utility and associated limitations.

Ionizing radiation (IR) promotes primary effects on DNA structure by directly inducing DNA strand breaks, particularly double-strand breaks (DSBs), and indirectly induces secondary effects by ionizing water molecules to produce reactive oxygen species (ROS) [2,3]. ROS oxidize lipids and proteins and also induce various forms of DNA damage, including the generation of oxidized bases and single-strand breaks (SSBs) in the radiated cells/tissue, ultimately leading to cellular apoptosis [4]. Furthermore, IR also promotes clustered DNA damage and induces covalent inter- and intra-strand cross-linking [5]. Additionally, there is a growing recognition of the immune response generated by RT. This review will focus on the implications of the innate myeloid and lymphoid lineages in anti-tumorigenic processes induced by RT and the use of various types of RT methods in cancer treatment [6,7]. Additionally, we also explore key strategies for enhancing the efficacy of RT while maintaining innate immunity during cancer therapy.

RT is an essential component of personalized medicine. It promotes immune suppression by inducing toxicity in bone marrow cells [8] and peripheral blood lymphocytes [9]. Additionally, RT activates innate immune systems and leads to bystander effects [10]. In the peripheral blood, dendritic cells, macrophages, and NK cells play a crucial role in regulating innate immunity and determining the efficacy of radiation therapy. In this review, we also summarize the non-ionizing radiation-induced biological responses and how the human system responses to radiation via endogenous antioxidants. To minimize normal tissue toxicity and protect innate immune cells, we discuss the molecular level activation of different types of innate immune cells within the innate immune systems following radiation exposure and explore potential strategies to enhance the efficacy of RT while preserving the integrity of the innate immune systems.

## 2. Radiation Therapy

RT delivers radiation in a fractionated regime. The cell-killing potential of radiation is influenced by factors such as linear energy transfer (LET), total dose and fractionation rate, and the radio sensitivity of the targeted tissues. Low LET radiation deposits a smaller amount of energy, whereas high LET radiation deposits a higher amount of energy in the targeted areas. The ultimate goal of RT is to deliver the optimum dose to tumor cells while minimizing exposure to healthy normal cells. Different types of RTs, such as X-rays, photons, and carbon ions, are available, and the decision to employ a particular type of radiation treatment depends on various factors, including tumor volume, tumor location, the sensitivity of the tumor as well as the surrounding normal tissue. RT is broadly categorized into two types [11] (1) External beam radiation (teletherapy) [11,12] and (2) Internal beam radiation therapy [11] (Figure 1).

### 2.1. External Beam Radiation Therapy

External beam RT (EBRT) [13] is a non-invasive method of using high-energy rays, such as X-rays, photons, protons, or particle radiation, to destroy tumor cells. EBRT is one of the standard treatment options for various types of cancers, including blood and skin cancers. EBRT uses high doses of radiation to destroy radio-resistance cancer stem cells (CSCs) when compared to non-CSCs [14]. In contrast, internal RT involves the use of radioactive materials inside the body/tumor. Radiation damages cancer cells through direct ionization of DNA [15] and other cellular targets via indirect effects [16,17] and the production of hydroxyl free radicals/reactive oxygen species (ROS) that induce DNA damage/strand breaks [18]. IR causes both single-strand breaks (SSBs) [2,19] and double-strand DNA breaks (DSBs) [4,20,21,22,23] highly occurring in proliferating tumor cells, and the resulting DNA damage often leads to cell death via apoptosis or necrosis (Figure 2 and Figure 3). There are several types of EBRT, including intensity-modulated RT (IMRT), volumetric-modulated arc therapy (VMAT), image-guided radiotherapy (IGRT), 4-dimensional radiotherapy, stereotactic radiotherapy, whole body irradiation and proton beam therapy [24]. IMRT is an advanced method of high-precision RT in which computer-aided linear accelerators are used to deliver précised doses of radiation to the tumor in the specified area. IMRT treatment plans use advanced technology to modify the intensity of each photon beam via dose-rate alterations and field modulation with multileaf collimators (MLCs) [25,26,27,28], thereby offering improved tumor coverage and reduced damage to health tissue associated with conventional RT.

IMRT causes early side effects such as hair loss, difficulty in swallowing, digestion problems, diarrhea, vomiting, headaches and bladder function impairments and late side effects including memory-related issues, spinal cord, lung, colon, infertility, and joint problems. To overcome these challenges, a range of motion management and mitigation strategies have been developed, including the use of 4-dimensional RT (4DRT) [29] to better understand tumor motion during breathing cycles and deliver more accurate and précised doses to the tumor. An innovative radiation treatment method called Stereotactic Ablative Radiotherapy [30] (SABR) delivers an intense dose of radiation to treatment targets in fewer fractions (1–5), at a higher dose per fraction (>5 Gy), when compared to conventional fractionation radiotherapy. SABR is also referred to as Stereotactic radiosurgery (SRS) for treating brain lesions and stereotactic body RT (SBRT) for extracranial tumors.

SABR plays a crucial role in treating many cancers, including non-small cell lung cancer (NSCLC), pancreatic cancer, liver cancer, and brain tumors, due to its remarkable local control and tolerance profile. SBRT can be delivered using IMRT or VMAT approach [31]. Additionally, the abscopal effect was observed when SABR was combined with immunotherapy [32]. VMAT treatment uses photons (X-rays) produced by a medical linear accelerator. The photons are rotated 360 degrees around the tumor, enabling a three-dimensional effect. VMAT promotes similar side effects as reported in IMRT. 4DRT is a 4-dimensional RT called respiratory gating; this kind of RT is used to treat cancers/tumors that move with the patient’s respiration, including lung, pancreatic and GI cancers. Superficial radiation therapy (SRT) delivers radiation from many different angles, enabling the delivery of higher radiation specifically to the tumor, thus sparing the surrounding normal tissue and producing fewer side effects.

In whole-body radiation (most commonly used for blood cancer [33]), the whole body is exposed to IR. Animal models have been used to test the effects of whole-body radiation [34]. Whole-body radiation significantly spares non-target organs, including the kidney and lungs [35], while eliminating diseased bone marrow. Proton therapy also represents external beam radiation and has several advantages over conventional X-ray (proton therapy) [36,37]. This advanced RT permits precise tumor targeting and sparing of surrounding normal tissue. Proton therapy is a painless treatment; the patient can return home and perform daily activities. However, the proton beams promote unavoidable sore, reddened skin [38,39] (called erythema/flushing), hair loss and low body energy.

### 2.2. Internal Beam Radiation or Brachytherapy

Internal beam radiation, brachytherapy [40], is another form of RT in which radioactive material is placed inside the patient’s body. There are three types of brachytherapy: (1) interstitial brachytherapy, where the radiation source is placed within the cancer tissue [41], e.g., prostate cancer; (2) intracavity brachytherapy, where the radiation source is placed in a body cavity (e.g., cervical, or endometrial cancer); and (3) episcleral brachytherapy, in which the radiation source is attached to the eye and is used to treat melanoma of the eye. Brachytherapy implants include: (1) Low-dose rate (LDR) implants, where the radiation source stays in a place for 1–7 days and is used to treat localized cancer, e.g., prostate cancer. (2) High-dose rate (HDR) implants, where the radiation source stays in place for 10–20 min to 2–5 days and is used to treat gynecological cancers. (3) Permanent implants, where an I-125 implant is placed permanently to treat prostate cancer. Other forms of radiation include the use of radioactive iodine (^131^I) to treat thyroid cancer and hyperthyroidism (Figure 1). ^131^I is employed in places where the cancer is already spread and cannot be removed by surgery. When ^131^I is taken as a tablet or oral liquid, ^131^I is absorbed by thyroid cancer cells or malignant pheochromocytoma or paraganglioma [42,43,44]; it destroys them by emitting gamma and beta particles (Figure 1). Similarly, ^32^P (which has a half-life of 14.3 days) and radium-223 (which has a half-life of 11.4 days) are used to treat prostate cancer [45].

Other radioisotopes used in clinics to treat cancers include Ga-68-DOTATATE, Lutetium-177-DOTATATE (to treat leukemia and pancreatic cancer), Ga-68-PSMA (to treat PCa) [46,47], and Lutetium-177-PSMA (also used to treat PCa) [48,49]. Samarium (153 Sm) lexidronam, which emits β-particles and has a half-life of 46.3 h, is used to treat bone cancer and bone metastasis [50] and radium 223 (^223^Ra). Ra 223 has a half-life of 11.4 days, used to treat metastatic castration-resistant prostate cancer (mCRPC) patients [51,52,53].

## 3. Non-Ionizing Radiation Induces Oxidative Stress

Exposure to non-ionizing radiation has been observed to stimulate an increase in the production of free radicals within the cellular environment. Many studies have suggested that non-ionizing radiation may trigger the formation of ROS in exposed cells in vitro [54,55,56,57] and in vivo [58,59,60]. The initial generation of ROS production in the presence of radiofrequency is regulated by the NADPH oxidase enzyme located in the plasma membrane. Consequently, ROS activates matrix metalloproteases, initiating intracellular signaling that communicates the presence of external stimuli to the nucleus. These alterations in transcription and protein expression become apparent following exposure to radiofrequency [61]. In a study by Kazemi et al., the impact of 900-MHz exposure on the induction of oxidative stress and the intracellular ROS levels in human mononuclear cells was investigated. Excessive elevation of ROS levels is a significant contribution to oxidative damage in lipids, proteins, and nucleic acids. A study conducted by Sepehrimanesh et al. demonstrated that RF-EMF leads to an increase in testicular proteins among adults, which is associated with a higher risk of carcinogenesis and reproductive damage [62]. Eroglu et al. reported that exposure to cell phone radiation decreases sperm motility and causes changes in sperm morphology. Similarly, microwave radiation has shown positive, negative and neutral effects in biological systems [63].

Goldhaber et al. documented a significant rise in fetal abnormalities and spontaneous abortions in pregnant women exposed to non-ionizing radiation [64]. Many of these effects are likely due to hormonal fluctuations [65,66]. To counteract the damage caused by ROS, living organisms possess’ anti-oxidative mechanisms, such as glutathione peroxidase, catalase and superoxide dismutase [67]. These mechanisms work by suppressing or impeding the chain reaction initiated by ROS, glutathione peroxidase, catalase, and superoxide dismutase. These defense mechanisms work by suppressing or impeding the chain reaction initiated by ROS. However, with excessive ROS production in response to non-ionizing radiation, the antioxidant defense mechanisms become impaired, leading to oxidative stress [68,69].

## 4. Innate Immunity

The immune system is classified into two categories: innate and adaptive immunity. Innate immunity is largely composed of myeloid/macrophages, natural killer (NK) cells, and dendritic cells. The innate immune system constitutes the first line of defense against invading microbial pathogens and recognizes the pathogens through pattern recognition receptors (PPRs) [70,71,72,73,74]. PPRs can detect conserved structures on pathogens termed pathogen-associated molecular patterns (PAMPs) [71]. However, recent findings suggest that the induction of immune effectors also commonly occurs in the absence of pathogen infection, which is termed sterile inflammation. Sterile inflammation is commonly found in RT-induced innate immune responses. PPRs also detect Damage Associated Molecular Patterns (DAMPs) [75,76] that originate within the damaged cell itself. The innate immune system initiates an immune response following the detection of DAMPs, which signals the status of tissue or cell damage or danger events. Innate immunity is activated by antigens and different immune cells, including dendritic, mast, natural killer (NK) cells, macrophages, monocytes, and granulocytes, to maintain the immune system. Adaptive immunity is mediated by lymphocytes such as T and B cells and is characterized by immunological memory cells that allow a long-lasting response. The effect of RT on adaptive immunity has been extensively discussed in the literature.

### Role of RT in Priming the Innate Immune Response

RT induces apoptosis, which triggers DAMPs. Examples of DAMPs include the extracellular release of high mobility group box1, production of cytokines such as type I interferon (IFN-1), release of nuclear (nDNA) and mitochondrial DNA (mtDNA) to cytoplasm, and production of reactive oxygen species (ROS) or free radicles (Figure 3). These signals induce a series of chemical and immunological reactions that affect immunity. Mitochondria contain numerous potent immunostimulatory DAMPs, including mitochondrial DNA (mtDNA), ATP [77] and ROS. Mitochondrial DAMPs engage the innate immune macrophages or neutrophils [78] upon release to the cytosol or into the extracellular environment. The release of mtDNA into the cytosol activates PPRs to trigger a variety of innate immune responses. One such PPR is the DNA sensor cyclic GMP-AMP (cGAM) synthase, which binds cytosolic double-strand DNA (dsDNA) derived from mitochondria. This results in the generation of the second messenger cGAMP and activates the cGAS-STING pathway at the endoplasmic reticulum, leading to the recruitment of the tank binding kinase 1 (TBK1) and activation of the IFN signaling pathway [79]. Another DAMP is mitochondrial ATP, the key transporter of chemical energy. Recently, in several models, it has been shown that IR causes the release of ATP from tumor cells and activates DC cells [80]. ATP binds to P2X7 on DC cells, leading to the activation of NLRF3 inflammasomes [81].

## 5. Mechanisms of Radiation-Induced Innate Immune Cell Activation

### 5.1. Dendritic Cells

Myeloid cells constitute a highly diverse population that is comprised mainly of dendric cells (DCs), monocytes and macrophages [82]. Dendritic cells play a crucial role in host immunity by promoting innate inflammatory responses to environmental or damage stimuli [83]. TNF-α and IL-1β are proinflammatory signaling molecules that are upregulated in response to IR. These molecules, in turn, activate antigen-presenting innate immune cells, including dendritic cells [84,85,86,87,88]. Dendritic cells are specialized antigen-presenting cells that play a crucial role in T-cell activation following RT-induced damage in tumor cells. Dendritic cells recognize DAMPs via specific receptors and matured dendritic cells [89] stimulate cytotoxic CD8+ T cells by antigen presentation and release of activating cytokines, thereby enhancing RT treatments. The intensity of radiation doses and the number of doses determines the immunogenic action of dendritic cells in RT. For example, repeated low radiation doses in a murine mammary carcinoma model create cytosolic DNA in tumor cells, activating the cGAS-STING pathway and the release of IFN-γ and subsequent T-cell activation [72,90,91] (Figure 4A,B). RT sensitivity depends in part on DNA exonuclease called 3′ repair exonuclease 1 (Trex1). Trex1 cleaves the RT-induced cytosolic DNA, thereby abrogating IFN-β production through the cGAS-STING pathway. The Trex1 level does not increase in response to multiple smaller fractions of radiation (8 Gy, three times); rather, it induces greater IFN-β production and activation of Bat3-dependent dendritic cells, leading to enhanced T cell responses. Compared to a single fraction of high-dose radiation, the induction of Trex1 in multi-low-dose RT is efficient and suggests that a fractionated low-dose of RT likely plays a role in enhancing immunogenicity [90,92].

### 5.2. Natural Killer Cells

NK cells effector lymphocytes play a crucial role in regulating innate immune responses, combating microbial infections, and controlling cancer. While IR has been shown to have a significant impact on NK cells, the underlying mechanisms behind this effect remain unclear [93]. NK cells are innate immune lymphocytes that can destroy target tumor cells by producing toxic and immunoregulatory cytokines [94,95]. IR has a significant effect on modifying NK cells. Previous studies have demonstrated that IR enhances the immune response by augmenting the antigenicity and adjuvanticity of malignant cells and by interacting with the tumor microenvironment (TME) [96]. Low-dose ionizing radiation activates NK cell functions, while high-dose ionizing radiation particularly impairs NK cell function (Figure 4A,B); however, this impairment can be reversed by interleukin-2 (IL-2) pretreatment [97,98,99]. Low-dose ionizing radiation at 75–150 mGy increases the secretion of NK cell effector proteins, such as IFN-γ and TNF-α [93,100]. Similarly, low-dose total-body irradiation (0.1 or 0.2 Gy X-ray) results in the suppression of experimental tumor metastases along with the stimulation of NK cell cytolytic functions in tumor-bearing rates [101,102].

Studies have identified that low-dose ionizing RT (LDIRT) can increase the immune response in vivo [100] with IFN-γ and TNF-α in the cultured medium of NK cells in response to LDIRT, and in addition, the P38 inhibitor (SB203580) drastically suppressing the NK cell cytotoxicity, cytokine levels, FasL and perforin [100,103]. Ames et al. (2015) identified that ex vivo NK cells are activated following low dose IL-2 and IL-15 and presented an increased ability to mitigate solid tumor cells in vitro and in vivo following RT [104]. A similar study reported that the presence of the cytotoxic effect in NK cells was boosted following RT in canine models of sarcoma, and the results from a clinical are progressing with possible abscopal effects. In general, NK cells produce perforin (Prf1) and granzyme B (GzmB) and induces cancer cell apoptosis (Figure 4) [105,106]. It is also observed that dendritic cells (DC) activate NK cells and promote tumor cell apoptosis [107]. A recent study utilized a reverse translational approach and revealed that NK cells play a role in immune enhancement through a CXCL8/IL-8-dependent mechanism in response to radiation [108]. Furthermore, the study suggests that NF-κB and mTOR mediate the secretion of chemokines that facilitates the infiltration of NK cells into tumor cells. Additionally, the study highlights that higher doses of radiation promote the transfer of adoptive NK cells and improve tumor control [108].

### 5.3. Macrophages

Macrophages play a crucial role in various aspects of immunity, including infiltrating the TME. Macrophages belong to the myeloid family and originate from circulating bone marrow-derived monocyte precursors. Macrophages are highly plastic cells that undergo significant changes in their function depending on the environmental cues in the TME, exerting a dual function on tumorigenesis by either antagonizing the cytotoxic activity of immune cells or enhancing the antitumor responses (Figure 4A,B). Macrophages are classified into two different phenotypes, M1 and M2. M1 macrophages are called classically activated macrophages in response to pathogens and take part in the immune response. M2 macrophages are known as alternatively activated macrophages involved in wound repair and have an anti-inflammatory role. Following recruitment, the monocyte precursor cells differentiate into macrophages in the tissue. The matured macrophages polarize to functionally different phenotypes in response to microenvironmental challenges in TME in tumor cells. Tumor-associated macrophages (TAMs), a major stromal component of TME, resemble the M2-polarized macrophages [109,110,111]. M1 macrophages are also involved in antitumor immunity, while M2 macrophages exert pro-tumorigenic activities.

Macrophages are recruited to the damaged or injured sites following the radiation exposure, where they carry out their phagocytic function [112,113,114]. Macrophage responses to RT range from promoting tumor growth to enhancing the immunogenic response, depending on the tumor type, environment, IR and dose, and fractionation. Inducible nitric oxide synthase (iNOS)+ M1-like macrophages undergo differentiation in response to local low-dose ionizing radiation, allowing the recruitment of tumor-specific T-cells and tumor regression in human pancreatic carcinomas [110,115]. Irradiated cells induce cytokine secretion and hypoxia within the damaged tissue, and the activation of the transcription factor HIF1-α has been shown to contribute to the recruitment of macrophages towards the immunosuppressive phenotype. Activated macrophages can directly destroy cancer cells by enhancing the phagocytosis of tumor cells through antibody-dependent cellular cytotoxicity. Alternatively, by secreting toxic/harmful molecules such as cytokines or tumor necrosis factors TNF or nitric oxide and promote cytolysis of cancer cells. The indirect killing of tumor cells involves the recruitment of immune cells, such as cytotoxic T-cells (Figure 4) [116,117,118].

## 6. Enhancing the Radiotherapy Efficacy through Microbubble Oxygen Delivery

Most solid tumors are characterized by hypoxia, which has a significant impact on how the patients respond to radiotherapy, chemotherapy, and immunotherapy. In fact, tumor hypoxia is a crucial factor in malignant progression, innate and adoptive immune escape [119,120]. Molecular oxygen has been shown to be an effective radio sensitizer that enhances radiation-induced DNA damage in tumor cells [121]. Compared to hypoxia, oxygen-rich conditions result in greater radiation-induced DNA damage within the tumor cells. Ionizing radiation generates free electrons by ionizing the tissue, and these electrons can cause IR-induced damage by producing highly reactive hydroxyl and hydrogen radicals. These short-lived radicals can damage macromolecules. However, in hypoxia, the extent of these reactions is limited due to the radicals’ instability. This phenomenon, also known as the “oxygen effect”, is the foundation of our understanding of how oxygen influences radio sensitivity.

Previous research has established that improving tumor oxygenation can enhance the efficacy of radiation treatment. For instance, preclinical mouse tumor models have shown that administering oxygen microbubbles (OMBs) sensitizes tumors to RT and limits tumor burden. Studies have indicated that intravenous injection of OMBs combined with ultrasound delivery of oxygen is more effective than intra-tumoral injection. Intravenous delivery of OMBs using contrast-enhanced ultrasound (CEUS) achieves true theragnostic functionality, minimally invasive, without causing infections, without displacing cancer cells or inducing injury. In one study, microbubbles and ultrasound waves were found to improve the efficacy of RT for breast cancer in mice [122,123]. Another study demonstrated that a new method involving the emulsion freezing-drying technique of microbubble oxygen delivery combined with iodine-125 brachytherapy increased the radiation sensitivity of the tumor cells [124]. However, despite the positive effect of microbubble oxygen cavitation on tumor response to radiation, hypoxia remains a limiting factor in brachytherapy.

Hypoxia is a significant characteristic of the TME in most solid tumors [125,126,127,128] and is closely linked with the immune response to tumors. The TME is a heterogenous mix of tumor cells, stromal cells, and immune cells that play a pivotal role in immune evasion. [129,130]. Hypoxic conditions in the TME activate multiple signaling pathways that reshape the immune system, leading to immunosuppression. For instance, hypoxia has been shown to affect the function of T-cells, tumor-infiltrating lymphocytes (TILs) and other immune cells, including CD8+ T-cells, NK cells, and natural killer T (NKT) cells. [131,132]. Recent studies conducted in murine models suggest that administering fractionated low doses of radiation with oxygen microbubble delivery to cancer patients could improve radiation effectiveness and boost the innate immune cell population and survival in the TME [133,134]. Recent literature shows that the application of oxygen mimetics, which utilizes the chemical properties of molecular oxygen, enhances radiation efficacy while minimizing radiation damage. Oxygen mimetics are considered “true radiosensitizers” [135,136]. Nitro-containing compounds and nitric oxide are used as oxygen mimetics [136].

## 7. Enhancing Radiation Efficiency and Maintaining Innate Immune System through the Combination of Microbubble Oxygen Delivery and Radioprotection with RT

Tumor cells tend to proliferate rapidly near the vascular supply leading to limited oxygen supply and the development of a hypoxic environment within the tumor. Consequently, necrosis ensues, marking the demise of these oxygen-deprived tumor cells [137]. It is well established that hypoxic tumor cells are more resistant to RT than aerobic tumor cells [138,139,140,141,142,143]. The tumor microenvironment (TME) is characteristic of hypoxia (low oxygen). The innate immune NK cells and dendritic cell survival and functions are controlled by the TME [144,145]. Oxygen is required to enhance the RT response; high doses of radiation are able to destroy tumor cells in the absence of molecular oxygen. However, these higher doses of radiation may also harm surrounding normal cells, including the innate immune cells that are the first line of defense. Lymphocytes are particularly sensitive to radiation. For example, RT leads to a decrease in the number of innate immune response lymphocytes in esophageal squamous cell carcinoma (ESCC) [146]. RT causes oxidative damage, and employing antioxidants such as superoxide dismutase, glutathione peroxidase, and catalase can protect innate immune cells from radiation-induced oxidative damage [147]. Therefore, it is important to use radioprotectors and microbubble oxygen delivery in cancer therapy to increase the efficacy of RT in cancer therapy and maintain the innate immune cell population at the tumor site (proposed model Figure 5). Radioprotectors are substances that are used to minimize DNA damage in normal and non-target tissues caused by ionizing RT. For these compounds to be effective, they must be present either before or at the time of radiation exposure. There are three categories of radioprotective agents: radioprotectors, adaptogens, and absorbents [137].

## 8. Potential Role of Endogenous Radioprotectors in DNA Damage and Immune Response to Radiation

Radioprotectors are naturally present in the human body and include both enzymatic and non-enzymatic substances that act as a potent antioxidant system [148]. The enzymatic systems, including superoxide dismutase, catalase, aldehyde dehydrogenase, and glutathione peroxidase, as well as non-enzymatic systems like antioxidant vitamins such as alpha-tocopherol and ascorbic acid, work together to eliminate the ROS from cells [148]. Catalases, found in peroxisomes adjacent to mitochondria, react with hydrogen peroxide to produce water and oxygen. Studies have shown that administration of SOD in mice protected the radiation-induced damage [149,150], and such studies possibly show that antioxidants play a role in restricting accidental radiation exposure and environmental radiation-induced damage.

Glutathione peroxidase reduces hydrogen peroxidase via glutathione. Additionally, peroxiredoxin, a family of sulfhydryl-dependent peroxidases, reduces hydrogen peroxide to water. Glutathione acts as a potent antioxidant by directly scavenging and neutralizing free radicals and ROS [151]. Glutathione also supports the activity of DNA repair enzymes and facilitates efficient mechanisms for DNA damage repair [152]. The glutathione pathway and its transcriptional regulator of NF-E2-related factor 2 (NRF2) promotes the survival, proliferation and function of T-cells, B-cells and macrophages by boosting cytokine production and regulating inflammatory responses [153].

The innate immune system plays a crucial role in protecting the body from environmental exposure to radiation. The innate immune system triggers an inflammatory response as a protective mechanism in response to radiation [154]. This causes cell damage and the release of danger signals, which activates macrophages and neutrophils. These immune cells migrate to the affected area and release cytokines and chemokines which promote inflammation. Inflammation initiates tissue repair [154]. The macrophages and dendritic cells recognize and remove the radiation-damaged cells through phagocytosis, preventing the accumulation of potentially harmful cells. NK cells detect the changes in the surface markers of damaged cells and induce cell death through cytotoxic molecules [155].

Currently, there are only two FDA-approved radioprotective compounds in RT: amifostine and palifermin [156]. Other radioprotectors, including antioxidants from food intake, work by neutralizing the free radicals that cause cellular damage induced by radiation. Typically, they are composed of sulfhydryl compounds. Adaptogens are plant-derived compounds that enhance the body’s resistance to radiation, while absorbents are chemicals that act as chelating agents to protect individuals from ingested radionuclides. Although the application of these compounds is effective, several side effects are also associated with their use. For example, synthetic radioprotectors such as 2-mercapto propionyglycine (2-MPG), cysteamine and S-(2-aminoethyl) isothiouronium bromide hydrobromide are very effective as radioprotectors. However, owing to toxicity and increased risk of side effects, these compounds are not approved for human application. Studies suggest that the application of steroids increased the radioprotection in mice and simultaneously increased innate immunity via NK cells [157,158,159,160,161]. Other studies have shown that the application of vitamin A with soybean oil, vitamin C, E and flavonoids offers radiation protection by suppressing oxidative stress [162,163,164,165,166].

## 9. Conclusions and Future Directions

RT is a vital component of cancer treatment, aiming to destroy cancer cells while minimizing damage to surrounding healthy tissues. The advent of advanced technologies has sparked a revolution in the field of RT, significantly advancing its potential for accurate and safe delivery of dose distribution to tumor sites. The development of sophisticated systems, including IMRT, SBRT and proton therapy, has revolutionized the field. IMRT is the most advanced form of 3D conformational RT, which allows for precise control over the intensity and direction of radiation beams, enabling highly targeted treatments and allowing for higher and more effective doses of radiation to be delivered without significantly affecting the surrounding healthy tissues. SBRT delivers high doses of radiation to tumors with accuracy while minimizing exposure to nearby healthy tissues. Proton therapy uses protons instead of X-rays, which allows more precise dose distribution and reduced damage to the surrounding healthy tissues. Additionally, the integration of imaging technology plays a significant role in enhancing RT. For instance, IGRT uses computed tomography (CT) or magnetic resonance imaging (MRI) to precisely locate and track tumors during treatment sessions. This real-time imaging allows for adjustments in radiation delivery and ensures optimal treatment accuracy.

RT has direct and indirect effects on tumor DNA resulting in DNA strand breaks and damage to tumor cells. Damaged cancer cells release DAMPs, which activate both innate and adoptive immunity. DAMPs primarily consist of protein molecules secreted by dying tumor cells. These DAMPs can serve as ligands for toll-like receptors (TLRs) expressed in the immune cells. They promote cytokine production, which in turn activates T-cells [167] and B-cells [168] and destroys cancer cells. The innate immune cells initiate an immune response following the detection of DAMPs, which signals the presence of tissue or cell damage or danger. Innate immune systems consist of various cell types, including macrophages, mast cells, basophils, eosinophils, monocytes, NK cells and dendritic cells. Among these, macrophages, dendritic cells, and NK cells play crucial roles in innate immunity. Innate immunity is activated by antigens and different immune cells, including dendritic, mast, natural killer (NK) cells, macrophages, monocytes, and granulocytes. Dendritic cells recognize DAMPs via specific receptors to stimulate cytotoxic CD8+ T cells by antigen presentation and release cytokines, thereby enhancing the efficacy of RT treatments. The strength and the number of doses determine the immunogenic action of dendritic cells in RT. For example, repeated low radiation doses in a murine mammary carcinoma model create cytosolic DNA in tumor cells, activating the cGAS-STING pathway, release of IFN-γ and subsequent T-cell activation [72,90,91].

Natural killer (NK) cells are innate immune lymphocytes that can destroy cancer cells by producing toxic and immunoregulatory cytokines [94,95]. Previous studies have demonstrated that IR has a significant effect on modifying NK cells. IR enhances the immune response by augmenting the antigenicity and adjuvanticity of malignant cells through interacting with the TME [96]. Low-dose IR activates NK functions, while high-dose IR particularly impairs NK cells. In response to local low-dose IR, Inducible nitric oxide synthase (iNOS)+ M1-like macrophages undergo differentiation, allowing the recruitment of tumor-specific T-cells and promoting tumor regression in human pancreatic carcinomas [110,115].

Adaptive immunity is mediated by lymphocytes such as T and B cells and is characterized by immunological memory cells that allow a long-lasting response. Scott et al. integrated the combination of advancements in RT and cancer genomics and proposed a novel algorithm for the genomic-adjusted radiation dose (GARD) to calculate the biological effect of the dose rather than the physical dose alone [169]. The GARD framework can be used as the new paradigm for the prescription of personalized radiation doses [170]. Tumor imaging and tumor genome-based determination of radiation dose will improve the accuracy of predicting radiation sensitivity to cancer cells, immune cells, and radiation efficacy, but these treatments need further exploration.

Most tumors exhibit hypoxic conditions, which serve as the rate-limiting factor for successful RT. Adequate oxygen levels are essential for optimal RT outcomes, and to address tumor hypoxia, oxygen-mimetic radiosensitizers such as metronidazole and misonidazole, can be utilized as hypoxic cell radiosensitizers [142,171,172,173,174,175]. Tumoral hypoxia is known to hinder the efficacy of RT and is inversely correlated with the patient’s survival rates [176]. Nonetheless, further clinical trials are necessary to explore the potential of oxygen mimetics as a substitute for molecular oxygen in RT. Selective gene targeting strategies have revealed that HIF regulates innate immune response mediated by macrophages, neutrophils, and dendritic cells. HIF1 represents a promising pharmacological target to enhance innate immunity [177].

A strategy to overcome the adverse effect of RT involves elevating the oxygen concentration at the tumor site. This approach also promotes the survival of innate immune cells. Recent advancements in cancer therapy have resulted in the development of oxygen-loaded microbubble delivery to the tumor sites, facilitating increased survival of innate immune cells and improved RT sensitivity [176,178,179,180]. In the future, image-guided RT, incorporating 360-degree rotation and the fusion of microbubble oxygen delivery with radioprotectors (antioxidants), holds the potential to bolster the innate immune system during ionizing RT. Several studies have demonstrated abscopal effects in patients treated with low and fractionated doses of RT [181,182]. However, the precise mechanism of abscopal is still under investigation, and future studies are needed to elucidate it fully. Recent studies conducted in murine models suggest that administering fractionated low doses of radiation with microbubble oxygen delivery to cancer patients could improve radiation effectiveness and boost the innate immune cell population and innate immune cell survival in the TME [133,134]. The innate immune NK cells and dendritic cell survival and functions are controlled by TME [144,145].

Recent advancements in cancer treatment utilizing non-ionizing radiation, such as electromagnetic fields (EMFs) combined with cisplatin, have shown promise as an alternative option [183,184,185,186]. However, the impact of non-ionizing radiation on innate immunity requires further exploration. To gain a better understanding of therapeutic options for enhancing innate immunity, additional clinical trials involving radiation in multiple tumor models are needed. Moreover, the current recruitment, ongoing and completed clinical trials (Table 1) at https://www.clinicaltrials.gov (accessed on 1 April 2023) lack sufficient data/results to draw conclusions regarding the response of intra-tumoral immunity to radiation. Therefore, large clinical trials are necessary to evaluate the role of the innate immune system in response to ionizing RT.

## Figures and Tables

**Figure 1 cancers-15-03972-f001:**
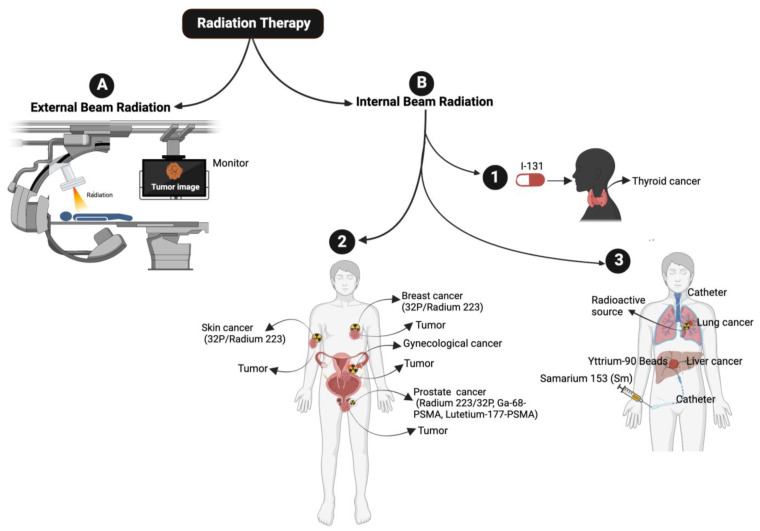
Overview of Different Types of Radiation Therapy: (**A**) The diagram depicts external beam radiation therapy. (**B 1**–**3**) The cartoon presents different models of internal beam radiation therapy: (**B 1**) Schematics show a thyroid patient being treated with radioactive iodine-131 medication. (**B 2**) Indicated radioactive capsules are embedded near or inside the tumor in breast, skin, gynecological and prostate cancer. (**B 3**) Cartoon shows the delivery of indicated radioactive beads/seeds in liver and lung cancers via a catheter.

**Figure 2 cancers-15-03972-f002:**
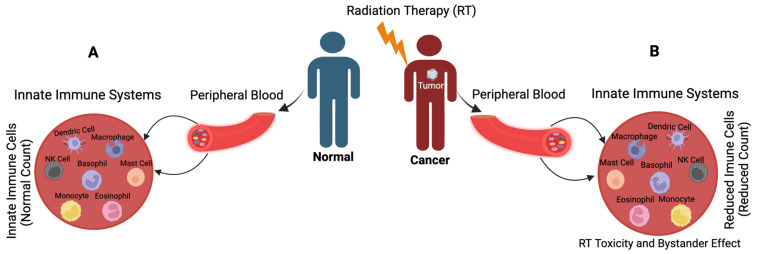
Presence of Innate immune systems in the peripheral blood. (**A**) Liquid biopsy shows the normal count of macrophage, mast cell, basophil, eosinophil, monocyte, NK cell and dendric cells in the peripheral blood. (**B**) Liquid biopsy from cancer patients treated with RT shows reduced counts of macrophage, mast cell, basophil, eosinophil, monocyte, NK cell and dendric cells in the peripheral blood in response to RT-induced toxicity and bystander effects.

**Figure 3 cancers-15-03972-f003:**
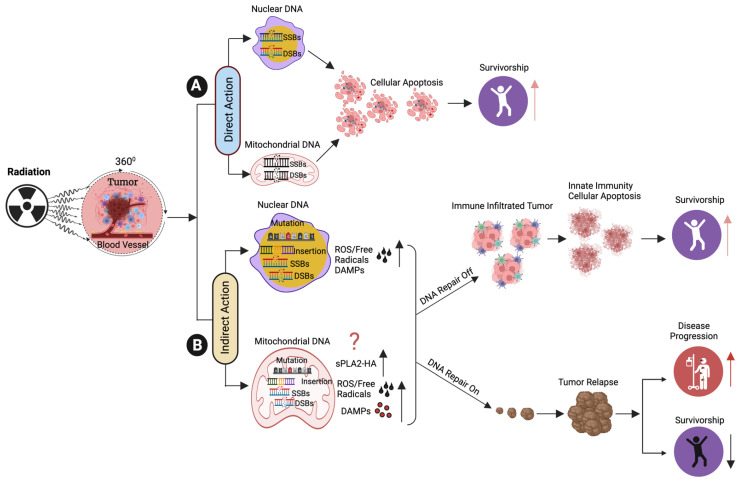
Direct and Indirect Effects of Radiation in Nuclear and Mitochondrial DNA from Tumor Cells. (**A**) The diagram presents the RT-induced apoptosis in tumor cells via direct single and double-strand breaks in nuclear and mitochondrial DNA. (**B**) The diagram represents the step-by-step RT-induced production of reactive oxygen species (ROS)/free radicles, which in turn promotes single and double-strand breaks in nuclear and mitochondrial DNA. While repaired single and double-strand DNA breaks can lead to tumor relapse or failure in DNA damage repair that can lead to cellular apoptosis and restricts disease-free survival among cancer patients, on the other hand, RT-induced apoptosis increases the disease-free survival rates among cancer patients. ? indicates may or may not happen, up-arrow indicates the increased survival and disease progression. The down arrow indicates poor survival.

**Figure 4 cancers-15-03972-f004:**
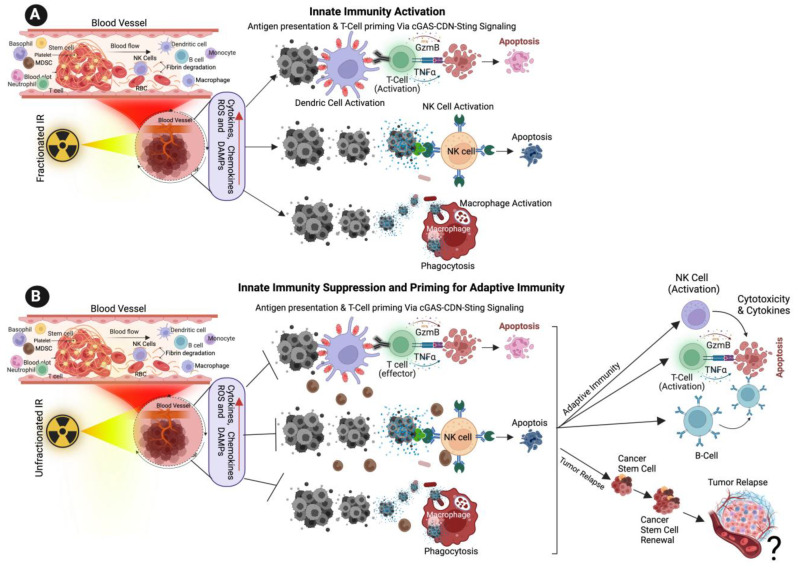
Effect of Ionizing Radiation on Innate Immune Activation. (**A**) The illustration shows the low dose of fractionated RT induces the production of ROS, cytokines, chemokines, and damage-associated molecular patterns (DAMPs) from cancer cells, DAMPs promoting the activation of innate immune dendritic cells, nature killing cells (NK cells) and macrophages. Activated dendric cells present tumor antigens and are primed for T-cell activation. Activated NK cells promote cytotoxic effects on tumor cells, and activated macrophages destroy cancer cells via active phagocytosis. (**B**) The cartoon illustration shows that a high dose of unfractionated radiation induces the production of ROS and DAMPs from cancer cells, inactivating innate immune dendritic cells, macrophages, and NK cells, promoting immune escape and tumor relapse. Unfractionated radiation also promotes adaptive immune cell activation via T-cells, NK cells and B-cells and promotes intra-tumoral immunity. ? indicates may or may not happen, arrow indicates the next step.

**Figure 5 cancers-15-03972-f005:**
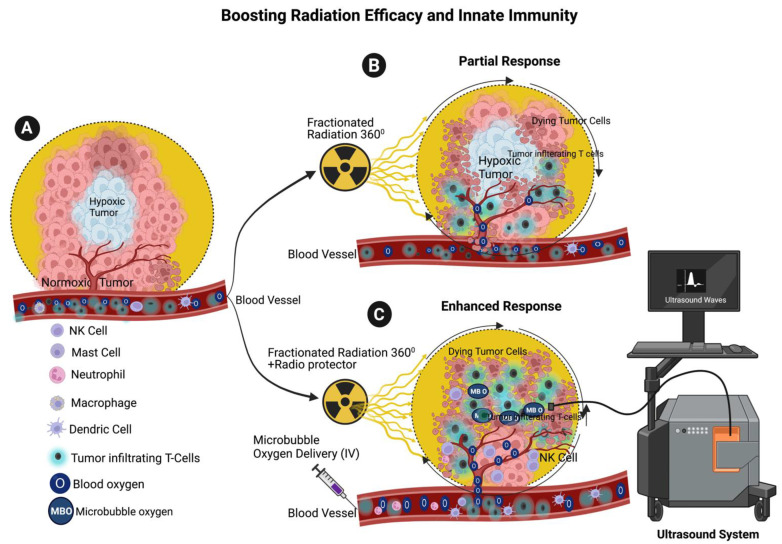
Proposed Therapeutic Option to Enhance the Efficacy of RT and Maintain the Innate Immunity. (**A**) The cartoon depicts tumor angiogenesis in response to indicated tumor microenvironment (TME) with all indicated immune cells. (**B**) Fractionated low doses of RT induce a partial response from tumors and innate immune systems. (**C**) The combination of both radioprotectors and microbubble oxygen with fractionated radiation promotes enhanced survival of immune systems, including macrophages, mast dendric, NK cells and tumor-infiltrating T-cell survival at the tumor site and increases the efficacy of radiation therapy.

**Table 1 cancers-15-03972-t001:** Ongoing, Completed, and Recruiting Clinical Trials Investigating the impact of RT on immune response at https://www.clinicaltrials.gov (accessed on 1 April 2023).

Trial Number	Study Title	Status	Cancer Type	Target Analysis
NCT02310594	Anti-tumor immune response in patients with cancer-undergoing radiation therapy	Completed (2022)	Malignant neoplasm	Innate & adaptive immune cells and serum markers.
NCT01376674	T-cell immunity during standard radiotherapy	Completed (2013)	Localized prostate cancer	Use of peripheral blood mononuclear cells
NCT01985958	A pilot study to evaluate the anti-tumor immunity in metastatic carcinoma of the pancreas.	Completed (2020)	Metastatic pancreatic cancer	Neutrophil, Platelets, Hemoglobin and Bilirubin
NCT05076500	Investigating the tumor immune response of radiotherapy	Recruiting currently	Cervical, rectal, Head and Neck cancer, nodal non-Hodgkin lymphoma	Immune signatures and immune phenotypes
NCT05035706	Anti-Leukemia immune response after irradiation of extramedullary tumors.	Recruiting currently	Leukemia	Lymphocytes
NCT01777802	Immune response in prostate, lung, breast and melanoma in response to SBRT and IMRT	Ongoing (Will be completed by 2023)	Melanoma, lung, prostate, and breast cancer	Change in circulating immune biomarkers and pro-inflammatory cytokine
NCT03383107	Effect of radiotherapy variables on circulating effectors of immune response and local microbiome	Completed (2021)	Prostate and breast cancer	Immune change before and after RT in and correlating with microbiome.
NCT05371132	Recruiting A study to evaluate CD8 PET imaging as a marker of immune response to stereotactic body radiation therapy (ELIXR)	Recruiting currently	Metastatic, malignant solid tumors	Monitor CD8+ T cells
NCT04624828	Immune response in evaluation in oligo-recurrent and oligo-progressive prostate cancer treated with SBRT	Recruiting currently	Prostate cancer	Monitor the dynamics of monocytes, granulocyte and NK cells
NCT03331367	Characterizing the immune response to prostate cancer	Completed (2020)	Prostate cancer	Immune markers from blood and urine

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
