# Peer review of "Innate Immune System in the Context of Radiation Therapy for Cancer"

_cancers, 2023, doi:10.3390/cancers15153972_

Round 1
Reviewer 1 Report (Previous Reviewer 1)
The authors responded to the previously indicated questions and the presentation of the manuscript has improved.
One thing to pay attention to, it seems that Fig. 2 is mentioned earlier than Fig. 1 in the text. In this case, a technical correction will be needed to mention the Figures in their order, i.e. Fig. 1 before Fig. 2
Reviewer 2 Report (Previous Reviewer 2)
I appreciate that the authors have addressed my comments and concerns in the revised version. This new draft seems with substantial improvements. I recommend accepting the paper for publication.
This manuscript is a resubmission of an earlier submission. The following is a list of the peer review reports and author responses from that submission.
Round 1
Reviewer 1 Report
The review article focuses on radiation therapy for cancer treatment and the activation of immune system cells in response to radiotherapy.
The article needs to be proofread and presented in a more consistent way. The Table needs to be aligned and proofread. A sharper introduction and conclusions would help to understand the purpose of this review article and how it contributes to the general knowledge in the research field and beyond.
Minor points.
1. Some paragraphs are too long. Although the text is clearly written and well-presented, making shorter and clearer paragraphs is beneficial for readers.
2. There are some typos and inconsistencies, including in association with references in the text, spaces, hyphens, etc.
3. Line 104. "...to treat thyroid..." Kindly check, it seems that the word is missing.
4. Line 128. "...and the complete system". Should it be the "complement system" instead?
5. Line 192. "promotes" is probably "promote".
6. Line 217. "by directly by producing" Kindly check
7. Line 259. "Radioprotectos" Kindly check
8. Lines 261-262. "Promotes necrotic" Should it be "promotes necrosis"?
9. Conclusions are not sharp enough. By reading the paper, it is unclear what new information it adds to the knowledge. Even though it is a review article, it could be clearly summarized in the Abstract, Introduction, and Conclusions, what is the purpose of this paper and what it brings to the readers.
10. Line 336. Abbreviations should be spelled, not just listed as it is "Abbreviations: PCa: RT; EBRT; IMRT, VMAT; IGRT; SSBs; DSBs; DAMPs; GzmB; TNF"
11. Table 1.
a) Kindly align the first line.
b) There are shades behind "Cancer Type", and "Malignant neoplasm" - please check and fix.
c) "Radiotherap" - kindly proofread the article and the table.
Reviewer 2 Report
Review on Innate Immune System in the Context of Radiation Therapy for Cancer
I have completed my review on manuscript Cancer-2336979, entitled, “Innate Immune System in the Context of Radiation Therapy for Cancer.”
Radiation therapy (RT) is an essential part of modern oncology care, and is frequently administered to cancer patients as part of their treatment plan. This article provides an overview of the various types of RT available for cancer treatment, and examines how they are applied. Additionally, the authors investigate the mechanisms by which innate immune cells in the immune system are activated in response to radiation exposure, and identify strategies to enhance the efficacy of radiation therapy while maintaining the integrity of the innate immune system.
The subject of review is very interesting and useful. Figure are professionally well prepared and understandable. Before making a positive decision, I have some concerns and comments about the present form of the manuscript that must be addressed first.
Comments for authors
Comment 1: In a review article, it is crucial to establish a strong foundation of background knowledge for readers to comprehend the complex subject matter. However, upon careful examination, it appears that the introduction section may lack sufficient information regarding the underlying mechanisms by which radiation impacts biological systems. I recommend the authors add information on this topic in the introduction and suggest consulting a recent article to expand their knowledge on the subject. This will help strengthen the introduction.
Article: Microwave Radiation and the Brain: Mechanisms, Current Status, and Future Prospects. International Journal of Molecular Sciences vol. 23 (2022). [https://doi.org/10.3390/ijms23169288].
Comment 2: Radiation therapy is already introduced in short form RT. Change Radiation therapy to RT in line 27, and cross-check the whole manuscript for such inaccuracies.
Comment 3: In line 139, Mitochondrial DAMPs engage the innate immune system upon release to the cytosol or into the extracellular environment. If authors mention which type of specific innate immune responses induces after RT is more helpful to readers.
Comment 4: In line 281 there are three categories of radioprotective agents: radioprotectors, adaptogens, and absorbents. These compounds are naturally also present inside the body in another form to protect the body from RT. If the authors add some details about how our body survives against environmental RT exposures and what our body's natural defense mechanism is against environmental RT as well as the role of the innate immune system that gives a more significant impact on readers.
Comment 5: In line 302 The damaged cancer cells then produce DAMPs that activate innate immune systems, which in turn prime T-cells to destroy cancer cells. If authors add some literature about B cells also here how B cells play a role in that because it’s one type of signaling pathway in which B cells also play an important role specifically B memory cells so it will give be more helpful for readers.
Comment 6: The authors discussed that only ionizing radiations cause ROS production and DNA damage. Latest research has revealed that non-ionizing nanosecond high-power microwave pulses possess the ability to induce ROS production, and DNA damage, and may hold significant potential for treating cancer. Given the importance of these findings, I encourage the authors to find literature on this and discuss such a topic in this review.
Comment 7: The paper contains errors and typos that make it difficult to understand and distort its intended meaning. I encourage authors to reread carefully and fix any grammatical errors.